# Biological and Immunological Characterization of a Functional L-HN Derivative of Botulinum Neurotoxin Serotype F

**DOI:** 10.3390/toxins15030200

**Published:** 2023-03-06

**Authors:** Zhiying Li, Bolin Li, Jiansheng Lu, Xuyang Liu, Xiao Tan, Rong Wang, Peng Du, Shuo Yu, Qing Xu, Xiaobin Pang, Yunzhou Yu, Zhixin Yang

**Affiliations:** 1Beijing Institute of Biotechnology, Beijing 100071, China; 2Pharmaceutical College, Henan University, Kaifeng 475001, China; 3Institute of Life Science and Biotechnology, Beijing Jiaotong University, Beijing 100044, China

**Keywords:** botulinum neurotoxin, functional molecule, FL-HN, immunological characterization, neurotoxicity, VAMP2

## Abstract

Botulinum neurotoxins (BoNTs) can cause nerve paralysis syndrome in mammals and other vertebrates. BoNTs are the most toxic biotoxins known and are classified as Class A biological warfare agents. BoNTs are mainly divided into seven serotypes A-G and new neurotoxins BoNT/H and BoNT/X, which have similar functions. BoNT proteins are 150 kDa polypeptide consisting of two chains and three domains: the light chain (L, catalytic domain, 50 kDa) and the heavy chain (H, 100 kDa), which can be divided into an N-terminal membrane translocation domain (HN, 50 kDa) and a C-terminal receptor binding domain (Hc, 50 kDa). In current study, we explored the immunoprotective efficacy of each functional molecule of BoNT/F and the biological characteristics of the light chain-heavy N-terminal domain (FL-HN). The two structure forms of FL-HN (i.e., FL-HN-SC: single chain FL-HN and FL-HN-DC: di-chain FL-HN) were developed and identified. FL-HN-SC could cleave the vesicle associated membrane protein 2 (VAMP2) substrate protein in vitro as FL-HN-DC or FL. While only FL-HN-DC had neurotoxicity and could enter neuro-2a cells to cleave VAMP2. Our results showed that the FL-HN-SC had a better immune protection effect than the Hc of BoNT/F (FHc), which indicated that L-HN-SC, as an antigen, provided the strongest protective effects against BoNT/F among all the tested functional molecules. Further in-depth research on the different molecular forms of FL-HN suggested that there were some important antibody epitopes at the L-HN junction of BoNT/F. Thus, FL-HN-SC could be used as a subunit vaccine to replace the FHc subunit vaccine and/or toxoid vaccine, and to develop antibody immune molecules targeting L and HN domains rather than the FHc domain. FL-HN-DC could be used as a new functional molecule to evaluate and explore the structure and activity of toxin molecules. Further exploration of the biological activity and molecular mechanism of the functional FL-HN or BoNT/F is warranted.

## 1. Introduction

Botulinum neurotoxins (BoNTs) are protein neurotoxins produced by anaerobic spore-forming clostridium Gram-positive bacteria (such as Clostridium botulinum, Clostridium butyricum, Clostridium barati, and Clostridium argentinum) [1,2]. BoNTs are the most toxic biotoxins, which can cause nerve paralysis syndrome in mammals and other vertebrates, and are classified as class A biological warfare agents by the US centers for disease control and prevention (CDC) [3,4]. According to the neutralization effects of specific antisera, BoNTs can be divided into seven traditional serotypes, A–G, and novel botulinum neurotoxins, including BoNT/H (previously considered to be a chimeric neurotoxin of BoNT/F and BoNT/A) and BoNT/X [5,6,7,8,9]. Among them, botulinum neurotoxin serotypes A, B, E, and F cause human poisoning, while BoNT/C and D cause neurotoxicity in animals [10]. The dose unit of botulinum neurotoxin poisoning is ng/kg, and the median lethal dose (LD_50_) of BoNT/F intraperitoneal injection in mice is about 2 ng/kg [3].

BoNT proteins are 150 kDa polypeptide consisting of two chains and three domains: the light chain (L domain, 50 kDa), a zinc-dependent metalloprotease that cleaves target proteins in neurons; the heavy chain (H, 100 kDa), divided into an N-terminal membrane translocation domain (HN, 50 kDa) and a C-terminal receptor binding domain (Hc, 50 kDa). The BoNT L and H chains are connected by interchain disulfide bonds, and BoNTs are activated after protease cleavage of the site of disulfide bond formation [11,12].

The poisoning process of BoNTs includes four stages: binding, internalization, transport, and paralysis [13]. The structure and poisoning process of BoNT/F are basically the same as those of other serotypes, while the mechanism of action and biological activity of BoNT/F are specific or different. The Hc domain mediates the endocytosis of the toxin into cells by binding to the ganglioside and synaptic vesicle glycoprotein 2C (SV2) dual receptors [1,14]. Endocytic vesicles induce conformational changes in BoNT/F in an acidic environment, and the HN domain of the heavy chain is inserted into the synaptic vesicle membrane to form a transmembrane channel to transport the L chain into the cell. After the L chain enters the cell, the disulfide bond connected to the H chain is reduced by the thioredoxin reductase-thioredoxin system (TrxR-Trx) and the L chain is released into the cytosol [12,15]. The L chain is a zinc-dependent protease with significant substrate specificity, which can specifically recognize and bind the soluble N-ethylmaleimide-sensitive factor attachment protein receptor (SNARE) proteins to exert its catalytic activity. The catalysis of BoNTs uses HEXXH as the signature motif on the L chain, which is conserved in all BoNTs. BoNT/F can complete the recognition and binding of the L chain to vesicle-associated membrane protein (VAMP) through a unique mode [16,17,18]. The L chain of BoNT/F cleaves VAMP and blocks the release of acetylcholine (ACh), resulting in nerve paralysis [19]. However, the process of BoNT poisoning involves a complex mechanism. The detailed research on the mode of action is insufficient [15,20], and some aspects, such as the function of HN, require further exploration.

BoNT has the characteristics of multiple serotypes and multiple subtypes, and the toxin-producing strains are complex and diverse. These phenomena bring great difficulties to the prevention of BoNT poisoning, and result in serious challenges for the biological defense against BoNT warfare agents. The toxoid vaccine, developed by the Michigan Department of Public Health (MDPH) in the 1970s, was estimated to produce antibodies in only 15% of subjects. The CDC discontinued the investigational pentavalent BoNT (ABCDE) vaccine for occupationally exposed persons in November 2011 because of the reduced immune efficacy [21,22]. BoNT/F can be grown and produced at low temperatures, and approximately 1% of natural botulism worldwide is caused by BoNT/F. BoNT/F poisoning mainly includes foodborne botulism and infantile botulism, which rapidly progress to paralysis and respiratory failure resulting from neuromuscular paralysis [23,24]. At present, there is no specific drug to treat BoNT poisoning. If the treatment of botulism is not timely, the patient needs long-term treatment and develops serious complications, which might lead to death. Therefore, prevention has become an important concern in BoNT poisoning, and there is an urgent need to develop efficient vaccines with strong protection, fewer adverse reactions, and convenient production and vaccination.

Vaccine development against BoNT has mainly focused on recombinant avirulent or attenuated subunit vaccines. Extensive research have shown that the Hc domain is the main target of BoNT recombinant subunit vaccines, and has protective antigenic properties [4,25,26,27,28,29]. Currently, the recombinant Hc subunit vaccines of BoNT/A, B, E, and F all have good protection against the biologically active BoNTs of their own serotype after immunization [30,31,32,33,34]. The L and HN domains of BoNTs have also been developed as recombinant subunit vaccines with strong potency to prevent animal botulism [30,35,36,37,38].

To define the function of the L-HN molecules of all BoNTs, we explored their biological and immunological characteristics [30,37,38]. Subsequently, in the present study, BoNT/F was selected to further explore the biological and immunological characterization of its functional molecules. The functional properties and biological activities of FL-HN were explored. We found that the di-chain FL-HN (FL-HN-DC) molecule had strong biological activity in vitro and low neurotoxicity in mice. Moreover, the immune protective efficacy of each functional molecule of BoNT/F as an antigen was assessed in immunized mice. The single chain FL-HN (FL-HN-SC) demonstrated the best immune protection effect among all molecules, which indicated that it could be used as a recombinant subunit vaccine against BoNT/F.

## 2. Results

### 2.1. Preparation and Identification of Recombinant Proteins

Based on the functional structure of BoNT/F (Appendix A), we constructed a series of BoNT/F functional molecules, and mutated the FL-HN functional molecule (FmL-HN). In addition, rVAMP2 was prepared using pTIG-Trx as an expression vector. Appendix A shows detailed information for each functional molecules of BoNT/F. These recombinant proteins were expressed in *E. coli*, purified by Ni-NTA affinity chromatography, and identified using SDS-PAGE (Figure 1a). Recombinant proteins rVAMP2 and FmL-HN were also identified by SDS-PAGE (Figure 1b,c), respectively. Each recombinant protein was further identified using Western blotting. FHc, FHc-C, and FHc-N could bind to anti-FHc horse sera antibodies (Figure 1d), FmL-HN, FL-HN, FL, FHN, and FH could bind to anti-BoNT/F horse sera antibodies (Figure 1e).

### 2.2. Biological Activity of FL-HN In Vitro

FL-HN and FmL-HN were identified by SDS-PAGE under reducing or non-reducing conditions after trypsin nicking, respectively (Figure 2a,b). The un-nicked proteins existed as single chains under both conditions, with a band size of 100 kDa. The nicked protein formed two bands of about 50 kDa after reduction, the size of them was consistent with that of FL and FHN. However, in the non-reducing environment, they were presented in the form of single chain, which revealed that FL-HN and FmL-HN were bivalent structures in which FL and FHN were connected by disulfide bonds after nicking.

Analysis of the enzymatic activity of recombinant FL-HN toward rVAMP2 in vitro showed that the cleavage efficiency of rVAMP2 by FL-HN-DC (Figure 2d) was higher than that of FL-HN-SC (Figure 2c). FL, as a positive control, could completely cleave the rVAMP2 protein into two bands of 15 kDa and 10 kDa (Figure 2e). No enzymatic activity was detected for FmL-HN-DC and FmL-HN-SC (Figure 2f).

### 2.3. Enzymatic Activity of FL-HN in Neuro-2a Cells

To further study the biological properties and functions of FL-HN at the cellular level, neuro-2a cells were selected as the cell model for related experiments. The molecules derived from BoNT/F were incubated with neuro-2a cells, and the activity of FL-HN was assessed by Western blotting, which could detect the content of cleaved rVAMP2 protein at the cellular level [36,37,38,39]. After co-incubating FL-HN with neuro-2a cells for 12 h, proteins in the cells were extracted for Western blotting detection. The signal intensity of VAMP2 detection in FL-HN-SC (Figure 3a), FL-HN-DC (Figure 3b), and FmL-HN-DC, and FmL-HN-SC (Figure 3c) treatment groups of cells was calculated, and the results showed that both FL-HN-SC and FL-HN-DC could enter the cells to cleave VAMP2, while FmL-HN-DC and FmL-HN-SC had no enzymatic activity toward VAMP2. As calculated using SPSS software, the EC_50_ of FL-HN-DC was 60 nM, which was about 23 times higher than that of FL-HN-SC, which had a very low cell-based activity (Figure 3d).

A cellular inhibition assay of FL-HN-DC activity in neuro-2a cells was performed with specific anti-FL-HN serum antibodies. The results showed that specific anti-FL-HN sera antibodies could effectively inhibit about 60% of FL-HN-DC activity in neuro-2a cells (Appendix A), while non-specific serum antibodies could not inhibit FL-HN-DC activity. This indicated that FL-HN-DC activity could be used to screen FL-HN-specific neutralizing antibodies or toxin poisoning blockers targeting FL-HN of BoNT/F.

### 2.4. Specific Neurotoxicity of Recombinant FL-HN

The results of a mouse toxicity test on FL-HN and FmL-HN showed that no neurotoxicity was detected when FL-HN-SC, FmL-HN-SC, and FmL-HN-DC were i.p. injected into mice, respectively. All mice died when the injection dose of FL-HN-DC was 100 μg/mouse and 20 µg/mouse, and the survival rate in the 10 µg/mouse and 5 µg/mouse group was 75% and 100%, respectively. The LD_50_ of FL-HN-DC was calculated by SPSS software as 11.4 µg. FL-HN-DC without the receptor domain showed similar biological activity to BoNT/F, while its neurotoxicity to mice was significantly reduced.

### 2.5. Immunoprotective Efficacy of Each Functional Molecule as a Subunit Vaccine against BoNT/F

These recombinant proteins were used as antigens and mixed with aluminum adjuvant to prepare subunit vaccines. Here, single-chain FL-HN (FL-HN-SC) was used to assess its immunoprotective efficacy. The immunoprotective efficacies of these recombinant protein antigens were evaluated according to the resistance of the immunized mice to the toxin and the level of neutralizing antibody in their sera. As shown in Table 1, FH, FHc, and FL-HN-SC protected mice against 10^4^ × LD_50_ of BoNT/F challenge after two and three immunizations. FL, FHN, and FHc-N antigens induced only 20% immunological protection against 10^2^ × LD_50_ of BoNT/F after two immunizations. Although the protective effect was enhanced after three immunizations, they still showed no protective effect on 10^3^ × LD_50_ of BoNT/F. Mice in the FHc-C or FsL-HN-immunized groups were not protected against BoNT/F after two or three immunizations. Dose-dependent BoNT/F neutralizing antibody levels were observed after two immunizations, with FL-HN having the best effect, inducing a neutralizing titer that was 2~3 times higher than that of the FHc and FH groups. The booster immunization produced higher neutralizing antibody titers. The levels of serum neutralizing antibodies were very low in mice immunized with FHc-N, FHc-C, FL, or FsL-HN.

### 2.6. Immunoprotective Efficacy of Different Combination Vaccines Containing Two Functional Molecules

Table 2 shows the result for vaccines comprising two antigens from different molecules. The FHc-C+FHc-N combination group induced almost no protection against BoNT/F, whereas the other groups could prevent death from 10^4^ × LD_50_ of BoNT/F. Compared with FL or FHN single antigen immunization, the protective efficacy of FL+FHN combination was significantly improved, and showed a synergistic effect. The neutralizing titer of FL+FHN combination was 2 IU/mL, which was much higher than the neutralizing levels of <0.1 IU/mL when immunized with FL or FHN alone; however, it was quite different from that of FL-HN-SC. This indicated that FL-HN exerted a better immune protection effect when it maintained its structural integrity. In addition, the FHc-C+FHc-N combination group’s resistance against BoNT/F and sera neutralizing antibody levels were much lower than those of the FHc antigen, showing no synergistic effect, which indicated that subdomain functional epitopes were weak antigenic epitopes. In the combined immunization group, except for the FHc-C+FHc-N and FL+FHN groups, the neutralizing titers of the other three combination groups were all above 30 IU/mL, further indicating that FL-HN, FHc, and FH were strong protective immunogens.

### 2.7. Determination of Humoral Immune Response after Immunization with Recombinant Proteins as Subunit Vaccines

The antibody levels in the sera of mice immunized with these subunit vaccines were detected using ELISA (Figure 4). The sera from the two or three immunizations was diluted 1:100 and then two or four-fold diluted to detect the specific antibody levels. The specific antibody titers after two immunizations reached 10^4^ and were dose-dependent. After three immunizations, the humoral immune response to these recombinant proteins in mice was enhanced, and the specific antibody titer reached up to 10^6^. Each recombinant protein antigen was able to elicit a strong specific immune response; however, some protein antigens such as FL, FHN, FHc-C, and FHc-N did not provide a strong protective effect. This showed that specific-antigen binding antibodies could not be used as decisive factors to evaluate the efficacy of immune protection.

### 2.8. Immune Protection Efficacy of FL-HN with Different Molecular Forms as Subunit Vaccines

FL-HN-SC showed the strongest immunoprotective effect among all functional molecule antigens and FL-HN-DC had a stronger biological activity than that of FL-HN-SC. FmL-HN-SC and FmL-HN-DC had almost no enzymatic activity; however, the effect of FL-HN structural changes or active site mutations on the efficacy of immune protection was unclear. We explored the immunoprotective efficacies of the four molecular forms of FL-HN (i.e., FL-HN-SC, FL-HN-DC, FmL-HN-SC, and FmL-HN-DC) (Table 3). The four molecular forms of FL-HN showed good protective effects. The survival rate of mice challenged with 10^4^ × LD_50_ of BoNT/F after the second and third immunizations was 100%. The level of neutralizing antibodies increased significantly with the increase of the number of immunizations. The neutralizing ability of FL-HN-SC as an antigen was the strongest, which was two to three times higher than that of FmL-HN-SC. It was suggested that the FmL-HN molecule might have lost the neutralizing antibody epitope at the active site. The neutralizing antibody levels of FL-HN-DC and FmL-HN-DC molecules were lower than that of the two single-chain molecules, which indicated that the double-chain structure did not enhance their immune efficacy, but slightly reduced the neutralizing antibody titers. The above results suggested that the single-chain FL-HN molecule was a suitable protective antigen.

### 2.9. Dose-Dependent Immunoprotective Efficacy of Recombinant Subunit Vaccines

To define the most ideal target antigens of BoNT/F, we performed dose-dependent immunization with FL-HN-SC, FmL-HN-SC, and FHc antigens. The mice after one immunization and two immunizations were challenged with 10^2^, 10^3^, and 10^4^ × LD_50_ of BoNT/F, respectively, and the survival of the mice was recorded (Table 4). FL-HN-SC, FmL-HN-SC, and FHc showed a dose-dependent response to BoNT/F after the first immunization. The protection rate of FL-HN-SC with 62.5 ng in each mouse against 10^3^ × LD_50_ of BoNT/F challenge was 100%, FmL-HN-SC and FHc could protect mice completely against 10^3^ × LD_50_ of BoNT/F challenge when the immunization dosage was 1 µg. The results for mice challenged with 10^3^ × LD_50_ of BoNT/F were calculated by SPSS software, and the median effective protective dose (ED_50_) of FL-HN-SC was 43.9 ng, and those of FmL-HN-SC and FHc were 258.9 ng and 200 ng, respectively. FL-HN-SC, FmL-HN-SC, and FHc did not show a significant difference in protective effect after two immunizations, the 15.6 ng dosage group could protect mice against the challenge of 10^4^ × LD_50_ of BoNT/F. In terms of neutralizing antibodies, FL-HN-SC and FmL-HN-SC had a stronger neutralizing ability, and the efficacy value detected at the same dose was 2~6 times that of FHc. The results indicated that FL-HN-SC was a more effective target antigen for BoNT/F than FHc, and the mutation of the FL-HN-SC active site would affect its protective efficacy.

## 3. Discussion

BoNTs are non-infectious neurotoxins that can be absorbed through mucosal surfaces, eyes, and broken skin. Current treatments for botulism rely primarily on the heptavalent botulinum antitoxin (H-BAT) [39], which was approved by the U.S. Food and Drug Administration (FDA) on 22 March 2013, the trivalent botulinum antitoxin (BoNT/A, B, E) used in Europe, and the monovalent botulinum antitoxin used in China. However, the use of these equine antitoxins still requires a large amount of drug administration and mechanical ventilation in a timely manner, and is often accompanied by serious adverse reactions, and involves long-term treatment. Furthermore, the use of antitoxins cannot reverse the effects of the intracellular neurotoxin [4,40].

Botulinum neurotoxin poisoning can be effectively prevented by vaccination. The most commonly used human botulism vaccine is the formalin-inactivated pentavalent vaccine (PBT) for high-risk populations and military personnel in the United States, which protects against serotypes A–E. The CDC discontinued the use of PBT on 30 November 2011 because of its decreased immunogenicity, decreased product potency, and increased incidence of injection site-related adverse reactions [41]. The bivalent vaccine rBV A/B has been continuously optimized by the U.S. Army Medical Research Institute of Infectious Diseases (USAMRIID), showing tolerable immunogenicity, and has entered phase II clinical evaluation to provide occupational protection for botulism laboratory workers [4,25]. None of the above vaccines protect against BoNT/F, which can cause botulism in humans, and so far there is no marketed vaccine for botulism. Subunit vaccines for the prevention of botulinum neurotoxin poisoning are mainly based on the Hc domain of BoNT/A, B, and E as protective antigens, and there are a few studies on BoNT/F vaccines. Our team developed a tetravalent botulinum vaccine (TBV) with recombinant Hc domains of BoNT/A, B, E, and F, which has significant advantages compared with the toxoid vaccines [34].

In recent years, the L and/or HN domains of BoNTs have been proved to generate protective antibodies [35,36,42,43]. In our studies, we also found that the L-HN molecules of BoNT serotypes A, B, and E have strong immunoprotective effects [30,37,38] as per the reports of Shone et al. on the translocation and effector domains BoNT/A and B [44]. The L-HN molecule of BoNT/E has been shown to have better immunoprotective efficacy than its Hc domain [32], which indicated that the L-HN molecule is superior to toxoid vaccines in terms of production and immune protection efficacy. The immunogenicity of BoNT/F molecules other than Hc requires further research to define their protective efficacy and the synergistic effects of multiple epitopes. Therefore, we further evaluated the immunoprotective efficacy of each functional molecule of BoNT/F, and explored the biological properties of recombinant FL-HN.

In this study, we obtained the recombinant functional molecules of BoNT/F, which had strong antigenicity. Among them, the ability of FL-HN to bind anti-toxoid horse sera antibodies was stronger than that of FH and FHc, and the signal intensity of FH was stronger than that of FHc. However, FHc had better binding ability to anti-FHc horse sera. These results suggested that FL-HN might be more immunogenic than FHc on the whole toxin.

The traditional mechanism of botulinum neurotoxin intoxication proposes that only Hc is responsible for the specific recognition of binding neuronal cells, and HN is responsible for transmembrane transport. However, some studies have found that the translocation domains in BoNT/A and BoNT/B are also involved in the binding process [45,46]. In our study, we found that the EL-HN molecule, lacking the Hc domain, can also exert a little neurotoxicity in animals, indicating HN might play a receptor binding role in neurotoxicity, similar to Hc domain [38]. To explore the biological activity of FL-HN, we mutated HELIH (amino acids 227–231) of FL-HN into AALIA to construct and prepare FmL-HN. In this study, FL-HN-SC and FmL-HN-SC were nicked by trypsin to form double-chain structures FL-HN-DC and FmL-HN-DC, linked by disulfide bonds. They existed in the form of a single chain under non-reducing conditions, and were reduced to two chains of L and HN in the presence of a reducing agent. In an in vivo toxicity experiment, only FL-HN-DC showed neurotoxicity in mice, indicating that the formation of disulfide bonds was necessary for the toxicity of botulinum neurotoxin, and further indicated that the enzymatic activity of the L chain resides lied at the catalytic site HELIH (227–231). The level of cleavage of VAMP2 by FL has been demonstrated in a cell-free assay [47]. Using FL as a positive control, we detected the catalytic activity of FL-HN on VAMP2 at the protein level. The cleavage efficiency of FL-HN-DC was higher than that of FL-HN-SC, and the enzymatic activities of FmL-HN-SC and FmL-HN-DC were not detected.

In addition, both FL-HN-SC and FL-HN-DC could enter neuro-2a cells to cleave VAMP2, in which FL-HN-DC had better activity. This indicated that the presence of disulfide bonds greatly improved the efficiency of L chain activity. FL, FmL-HN-SC, and FmL-HN-DC were all unable to cleave VAMP2 protein. It was suggested that the HN structure is crucial for the enzymatic activity of FL-HN, and mutation of the catalytic site almost completely inhibited the cleavage ability of L chain toward VAMP2. FL-HN retained a similar structure and properties to the full-length toxin, without the assistance of the Hc domain. Moreover, the HN structure might have a structural epitope that could bind to neurons to assist the entry of the light chain into the cell, explaining the effect of FL-HN-DC on mouse toxicity. In the exploration of FL-HN-DC enzyme activity, we used FL-HN super-immune sera antibodies to try to block the cleavage effect of FL-HN-DC toward VAMP2 in cells, and found that specific antibodies could inhibit the enzyme activity of FL-HN-DC (Appendix A). This result proved that FL-HN had similar structure and function to L-HN of BoNT/A and B [45,46]. The strong enzymatic activity retained by FL-HN-DC was of great significance to further define the mechanism of botulinum neurotoxin poisoning and to screen specific antibodies or receptor binding blockers.

Recent reports showed that L-HN can still allow active proteases to enter the cytoplasmic matrix of target cells in the absence of the Hc structure [45,46]. Compared with the full-length toxin, the toxicity of L-HN of BoNT/A was greatly reduced, but its neurotoxicity showed that the L-HN molecule still had the potential to execute membrane translocation. The HN structure of BoNT/A could be combined with ganglioglycerides. Immunofluorescence detection using neuro-2a cells proved that the domain involved in binding to neuronal cells was not only the traditionally considered Hc domain [48], explaining that in the absence of Hc, L-HN still retained a biology activity similar to the natural toxin. Ayyar et al. expressed residues 729 to 845 of the HN of BoNT/A and demonstrated that this molecule bound to mouse brain SNAPs, substantially inhibited binding of the intact toxin, and revealed that HN729-845 could bind to the membrane of neuro-2a cells [35,46]. Research on HN structures of different subtypes of BoNT/A found that the toxicity and the efficiency of cleaving SNAP25 were affected by the HN structure [49]. The HN structure is critical in forming the channel for L-chain translocation and might be a determinant of the toxin’s potency. L-HN of BoNT/B also demonstrated its ability to cleave several VAMP substrates in vitro, and retained its native structure and function after deletion of the Hc domain. The double-stranded structure of EL-HN after trypsin nicking has neurotoxicity and can efficiently cleave SNAP25 in vitro, proving that EL-HN retains similar activity to BoNT/E [38]. Herein, the di-chain FL-HN (FL-HN-DC) showed neurotoxicity and could also enter neuro 2a cells to cleavage VAMP2. Meanwhile, the biological function and molecular mechanism of L-HN without Hc, such as the function of HN, require further exploration.

In the current study, the FL-HN-SC antigen showed better immunoprotective efficacy than FHc. Compared with FL or FHN single antigen immunization, the FL+FHN combination showed a significant synergistic effect; however, the protective effect and neutralizing antibody levels of the FL+FHN combination were markedly inferior to those of FL-HN. It was shown that the complete structure of the fusion connection of L and HN could induce higher serum neutralizing antibody levels and better protective efficacy, indicating that FL-HN, especially the junction of FL and FHN, might contain important neutralizing antibody epitopes.

Neutralizing antibodies can inhibit the binding of toxins to neuronal cell surface receptors, prevent toxins from transmembrane entry into cells, inhibit toxin transport across the cell membrane, or remove circulating toxins, which are important criteria for evaluating the efficacy of immune protection [50]. Protective antigens with strong protective potency are not only crucial to the research of BoNT/F vaccines, but also are significant for the preparation and screening of antibodies that specifically neutralize toxins. Our results on the immunoprotective efficacy of different molecular forms of FL-HN showed that all four molecular forms of FL-HN could had protective potency against BoNT/F. FL-HN-SC had the best ability to resist the toxin after one immunization. Although disulfide bonds allowed FL-HN-DC to retain the strong biological properties of BoNT/F, they did not enhance the neutralizing antibody level. FL-HN-SC was the molecular form with the best immunoprotective effect, which suggested that the L-HN junction and the mutation site of the L chain have certain antigen-neutralizing epitopes.

Previous studies on neutralizing antibodies of botulinum neurotoxins mainly focused on the Hc domain. In the study of neutralizing antibodies against the BoNT/F subtypes, it was found that the epitopes of multiple effective antibodies are located in L-HN and HN. The equimolar combination of 6F5.1+6F11+6F13, which binds HN effectively, can neutralize BoNT/F1, F2, F4, and F7, and the neutralizing potency is higher than that of the heptavalent botulinum antitoxin [51]. Human monoclonal antibodies with high affinity (1B18 or 4E17) bind to BoNT/A and B or BoNT/A, B, E, and F, respectively, and both antibodies bind to a relatively conserved epitope in HN [52]. Among the three combination monoclonal antibodies that can effectively neutralize four different BoNT/E subtypes, the epitope of the more potent 3E2 antibody is located at the junction of L and HN [53]. Epitopes of two of the four monoclonal combination antibodies capable of effectively neutralizing BoNT/C, BoNT/CD, BoNT/DC, and BoNT/D toxins are on HN and one is on L [54]. In the investigation of antibodies with binding ability to BoNT/H, the epitope for the monoclonal antibody showing activity is located in HN [55]. It was suggested that HN had some important epitopes. Therefore, these HN-specific antibodies could effectively bind the botulinum neurotoxin molecule, prevent cell translocation, or inhibit the release of the L chain to exert an antidote effect. The higher neurotoxicity of FL-HN-DC and the stronger immunoprotective efficacy of FL-HN-SC indicated that there are important antibody epitopes at the junction of L and HN of BoNT/F.

Single-domain variable heavy-chain (VHH) antibodies targeting the L domains of BoNT/A and BoNT/B can be used as antidotes or to investigate the structure and function of protease domains [56]. EL-targeted VHHs can effectively inhibit SNAP25 protease activity by intraneuronally delivering protease inhibitors through bio-molecular carriers [57]. The neurotoxicity and biological activity of FL-HN-DC could support further studies on the structure and function of L, HN, and L-HN junctions. Moreover, our results demonstrated that specific anti-FL-HN serum antibodies could effectively inhibit the FL-HN-DC activity in neuro-2a cells. This indicated that it would be significant to screen FL-HN-DC or light chain specific neutralizing antibodies or drug molecules, and explore the molecular mechanism of the biological activity of FL-HN-SC or FL-HN-DC.

## 4. Conclusions

The functional molecule FL-HN-SC had the highest protective efficacy, containing some important neutralizing epitopes at the junction of the L and HN domains and the enzymatically active site on the L chain. FL-HN-SC was the best immunogen for BoNT/F and could be used as a subunit vaccine to replace the FHc subunit vaccine and toxoid vaccine. In addition, FL-HN-DC retained the properties and activities of the neurotoxin and could enter cells to cleave substrates, and thus could be used as a tool for screening drugs and exploring molecular mechanisms. Therefore, these functional L-HN molecules provided an additional tool to investigate the structure function relationship or the degree of immune protection of different molecules of BoNT during intoxication. These findings also provided new strategies for developing vaccines and therapeutic antibodies against BoNT.

## 5. Materials and Methods

### 5.1. Animals and Ethics Statement

Female Balb/C and Chinese Kunming (KM) mice were randomly allocated to different groups as experimental animals, which were provided by Vital River Laboratory Animal Technology Co., Ltd. (Beijing, China). All mice were housed in pathogen-free conditions. All animal experiments were approved by the Animal Care and Use Committee of our institute (Beijing Institute of Biotechnology) and were conducted in accordance with the ethical guidelines and the National Institutes of Health guide for the care and use of Laboratory animals (IACUC-DWZX-2021-014).

### 5.2. Identification and Production of Recombinant Proteins

The genes encoding each molecule (FL, FHN, FL HN, FsL-HN, FH, FHc, FHc-N, FHc-C, FmL-HN, and VAMP2, Appendix A) were inserted into the prokaryotic expression vector pTIG-Trx. The recombinant plasmids encoding each functional molecule were identified by sequencing analysis. These pTIG-Trx prokaryotic expression plasmids encode the following BoNT/F molecules with a C-terminal His-tag: FL (residues 1–436), FHN (residues 437–857), FL-HN (residues 1–857), FsL-HN (residues 244–627), FH (residues 537–1278), FHc-N (residues 858–1075), FHc–C (residues 1016–1278), and FHc (residues 858–1278).

The recombinant plasmids were transformed into *E. coli* BL21 (DE3) competent cells for expression. A single colony was inoculated in 5 mL of 2 × YT medium and cultured at 37 °C with shaking at 220 rpm until the absorbance of the bacteria at 600 nm (OD_600_) reached 1.0. The cells were inoculated into 400 mL of 2 × YT medium containing 100 µg/mL ampicillin at 37 °C at 220 rpm until an OD_600_ of 1.0 was reached. Then, 0.4 mM isopropyl-β-D-galactosylthiopyranoside (IPTG; Promega, Madison, WI, USA) was added to induce the expression of the recombinant proteins. After overnight incubation at 16 °C, the cells were collected by centrifugation at 6000× *g* for 15 min, and resuspended in binding buffer (20 mM Tris-Cl, 500 mM NaCl, pH 8.0), and lysed by ultrasound in an ice water bath. The cells were centrifuged at 12, 000× *g* for 15 min at 4 °C to collect the supernatant. The supernatant was filtered using a 0.45 µM microporous membrane and stored at 4 °C for the next experiment.

AKTA Explorer (GE Healthcare, Piscataway, NJ, USA) was used for the chromatographic purification of the recombinant proteins. The recombinant protein in the supernatant was purified using a 5 mL HisTrapTM HP 5 (GE Healthcare Bio Sciences, Uppsala, Sweden) affinity column, and gradient elution was performed using imidazole. The eluted sample was collected, and the buffer of the final product was exchanged with phosphate-buffered saline (PBS, pH 7.4). The recombinant proteins were then identified using sodium dodecyl sulfate-polyacrylamide gel electrophoresis (SDS-PAGE).

The antigenicity of each recombinant protein was detected by Western blotting. After separation by 12% SDS-PAGE, these recombinant proteins were transferred to a polyvinylidene fluoride (PVDF) membrane and blocked with 5% skim milk at 37 °C for 2 h. The membrane was then incubated with anti-FHc (1:2000) [58] and anti-BoNT/F (1:1000) horse sera antibodies (from National Institutes of Food and Drug Control, Beijing, China) in 5% skimmed milk at room temperature for 1.5 h or overnight at 4 °C. After washing with TBS-T (Tris-buffered saline with 0.1% Tween-20), the membrane was incubated with 1:5000 diluted goat anti-horse IgG-horseradish peroxidase (HRP) (Santa Cruz Biotechnology Inc., Santa Cruz, CA, USA) at room temperature for 30 min. After washing with TBS-T, Western blotting solution (Thermo Fisher Scientific, Waltham, MA, USA) was added to the membrane, which was exposed on a gel imager (Bio-Rad, Hercules, CA, USA).

### 5.3. Specific Neurotoxicity Test of Recombinant FL-HN

FL-HN or FmL-HN was proteolytically cleaved using trypsin (NEB, Ipswich, MA, USA) (1:100 ratio of trypsin to L-HN) for 60 min at 37 °C. Then, 4 µg/mL trypsin inhibitor (Soybean, Sigma, St. Louis, MO, USA) was added to stop the digestion. Both the un-cleaved FL-HN (single chain, SC) and cleaved FL-HN (dichain, DC) with disulfide bonds were determined using SDS-PAGE.

The neurotoxicity of the FL-HN or FmL-HN in mice was determined using an LD_50_ assay. In FmL-HN, the HELIH (227-231) amino acids in the FL-HN functional molecule were mutated into AALIA (227-231). The un-nicked and nicked FL-HN or FmL-HN proteins were diluted in sterile normal saline. KM mice (specific pathogen free (SPF) grade females, 16–18 g) were injected intraperitoneally (i.p.) with a series of diluted FL-HN or FmL-HN, observed for seven days, and their survival was recorded. The lowest amount of toxin that killed 50% of mice was considered to be one minimal lethal dose (LD_50_) and was expressed as the number of LD_50_ units/mg of protein.

### 5.4. Enzyme Activity Detection of FL-HN In Vitro

Recombinant vesicle associated membrane protein 2 (rVAMP2) (aa 30–92) was successfully prepared and the proteolytic activity of FL-HN was assayed using the endopeptidase method. Briefly, assays were performed in a final volume of 50 µL TBS (Tris buffered saline) containing serially diluted FL-HN-SC, FL-HN-DC, or FL, where the molar ratio of the light chain to the substrate protein rVAMP2 was 0, 0.5, 1, 2, respectively. The concentration of rVAMP2 in each group was consistent (3 µM/L). Then, the mixed sample was incubated at 37 °C for 0.5 h to cleave rVAMP2. The reaction was carried out at 37 °C for 30 min and terminated by adding 4 × SDS-PAGE sample loading buffer. SDS-PAGE was then used to display the results.

### 5.5. Enzymatic Activity of the FL-HN in Neuro-2a Cells

The enzyme activities of FL-HN and FmL-HN were detected at the cellular level using Western blotting. Mouse neuro-2a cells (Procell Life Science &Technology Co., Ltd., Wuhan, China, Cat# CL-0168) were cultured in minimal essential medium (MEM, Grand Island, Gibco, NY, USA) containing 10% fetal bovine serum at 37 °C and 5% CO_2_ in an incubator (Thermo Fisher Scientific Inc., Waltham, MA, USA). The cells showing good growth conditions were used for the experiments. When the cell growth density reached 80%, 5 × 10^4^ cells per well were cultured in 12-well plates. After 24 h of culture, the medium was replaced with serum free medium (SFM, Gibco) for 12 h of starvation treatment. Then, different proportions of protein were added and culture was continued for 12 h. Cells were collected and protein was extracted. Thereafter, centrifugation was performed at 12, 000× *g* at 4 °C for 15 min, and the supernatant was collected for separation using 12% SDS-PAGE.

The separated proteins were transferred to PVDF membranes (Millipore Corp, Bedford, MA, USA) using a membrane transfer device and incubated with 5% skim milk for 2 h at room temperature. The membranes were then incubated with rabbit anti-rVAMP2 antibodies (Abmart Inc., Shanghai, China; 1:2000) and mouse anti-glyceraldehyde-3-phosphate dehydrogenase (GAPDH) antibodies (ZSGB-BIO, Beijing, China; 1:2000) overnight at 4 °C. The membranes were then washed three times with TBS-T), and incubated with 1:5000 diluted HRP-conjugated anti rabbit IgG (ZSGB BIO) and 1:5000 diluted goat anti-mouse IgG HRP (ZSGB BIO) at 37 °C for 45 min. After the membrane was washed three times with TBS-T, Western blotting solution (Thermo Fisher Scientific) was added to the membrane and exposed on a gel imager (Bio-Rad). The immunoreactive band corresponding to cleaved and uncleaved rVAMP2 were quantified by densitometry using Image J software (version 1.8.0, NIH, Bethesda, MD, USA).

### 5.6. Inhibition Assay of FL-HN-DC Activity via Anti-FL-HN Serum Antibodies

The anti-FL-HN serum antibodies (1:10) were incubated with FL-HN- DC (30 nM) at 37 °C for 30 min in cell culture medium. The non-specific and ant-FL sera antibodies were used as negative controls. Then, these mixtures were added to neuro 2a cells, culture was continued for 12 h, and the enzymatic activity of FL-HN-DC was determined as in Section 2.5. Finally, the medium was removed and the cells of each group were washed three times with PBS before being lysed. The supernatant of the lysate was collected for Western blotting analysis.

### 5.7. Immunization of Mice and Challenge with BoNT/F

SPF female Balb/c mice at 6~8 weeks old were randomly divided into different groups (*n* = 10 per group). Single antigen (1 or 10 µg/dose) and combined antigens (0.5 or 5 µg for each/dose) were diluted in sterile PBS, and formulated with 10% aluminum hydroxide gel (Alhydrogel; Brenntag Biosector, Frederikssund, Denmark). These mixtures were placed at room temperature for 1 h. The PBS group was used as the negative control. Mice were inoculated by intramuscular injection, and the injected volume of each mouse was 100 µL. The booster immunization was carried out every three weeks. Then, 200 µL blood samples from each mouse were collected at the third week after two or three vaccinations. The mice immunized twice and three times were challenged i.p. with 10^2^, 10^3^, 10^4^ × LD_50_ BoNT/F, which was assayed using the botulinum antitoxin standard (from National Institutes of Food and Drug Control, Beijing, China), and the survival of the mice was recorded for one week. These antigen proteins with strong immune protection effect were used for a dose-dependent protection experiment. Female Balb/c mice were immunized once or twice using serially diluted (4000 ng, 1000 ng, 250 ng, 62.5 ng, 15.6 ng, and 3.9 ng) antigen formulated with aluminum adjuvant. The mice were immunized once every three weeks. The mice immunized once and twice were challenged with 10^2^, 10^3^, 10^4^ × LD_50_ BoNT/F by i.p. injection, and the survival of the mice was recorded for one week. Euthanasia was then performed for the living animals. After the animal experiment, the body was stored in a −20 °C refrigerator, and a special treatment company carried out the unified treatment.

### 5.8. Determination of Neutralizing Antibody Titers in Mouse Serum

Neutralization antibody levels were detected with SPF grade 16–18g female KM mice. BoNT/F was diluted to 10^2^ × LD_50_/mL with diluent (50 mM KH_2_PO_4_, 50 mM Na_2_HPO_4_, 1M NaCL, 1% (*w*/*v*) gelatin, pH 6.5), and 1 mL of diluted toxin was added into each tube. The mouse sera of each group was diluted according to the protective effect, and then prepared according to a two or four-fold gradient. Different doses of sera were added in turn. The system was replenished to 2.5 mL with the diluent. The serum antibodies and toxin were mixed evenly, and incubated in a 37 °C incubator for 30 min to allow the toxin and antibody to fully react. Then, each mouse was injected i.p. at 500 µL/mice; the dose of toxin was 20 × LD_50_ in each mouse, and there were four mice in each subgroup. The survival of the mice was observed for a week after injection, once every day. The titer of serum neutralizing antibody was calculated in international units per milliliter (IU/mL) relative to the World Health Organization (WHO) BoNT/F antitoxin level.

### 5.9. Determination of the Humoral Immune Response after Immunization with Functional Molecule Antigens

The humoral immune responses of all functional molecule antigens were measured using an enzyme-linked immunosorbent assay (ELISA) according to a previously published method [37,38]. In short, ELISA plates were coated with 100 µL of recombinant protein antigens (2 µg/mL). Mouse serum samples were two or four-fold serially diluted from 1:100 using a 2% BSA blocking solution. The diluted samples (100 µL) were added to each well and incubated for 1.5 h at 37 °C. After washing with PBS-T six times, these plates were added with 100 µL of 1:5000 diluted IgG-HRP per well and incubated for 30 min at 37 °C. The samples were washed with PBS-T, followed by the addition of 50 µL of citrate buffer (pH 5.0) with 0.02% (*v*/*v*) hydrogen peroxide and 0.04% (*w*/*v*) o-phenylenediamine to each well. Finally, 50 µL of 2 M H_2_SO_4_ per well was used to stop the reaction, and the absorbance was read at 492 nm on a microplate reader. Serum samples of each mouse in all immunization groups was measured separately, and the geometric mean titer (GMT) for each group was calculated using the average values.

### 5.10. Data Analysis

The mean ± the standard deviation was used to express the quantitative data. Differences in antibody immune responses between groups were analyzed using one-way analysis of variance (ANOVA) or Student’s *t*-test. Fisher’s exact test was used to determine the statistical difference in survival between the treatment groups. Statistical significance was accepted at *p* < 0.05 for all data tested. Densitometric analysis was performed using Image J software (NIH, Bethesda, MD, USA) and nonlinear fitting was performed using GraphPad prism 8 (version 8.0.2.263, GraphPad Software, Inc., San Diego, CA, USA) Probability analyses with 95% confidence levels were used to determine the effective dose (ED_50_) to protect half of the mice in the dose-dependent immunization or the 50% effective concentrations (EC_50_) in the enzymatic activity of FL-HN in neuro-2a cells using IBM SPSS Statistics (version 19.0, IBM Corp., Armonk, NY, USA).

## Figures and Tables

**Figure 1 toxins-15-00200-f001:**
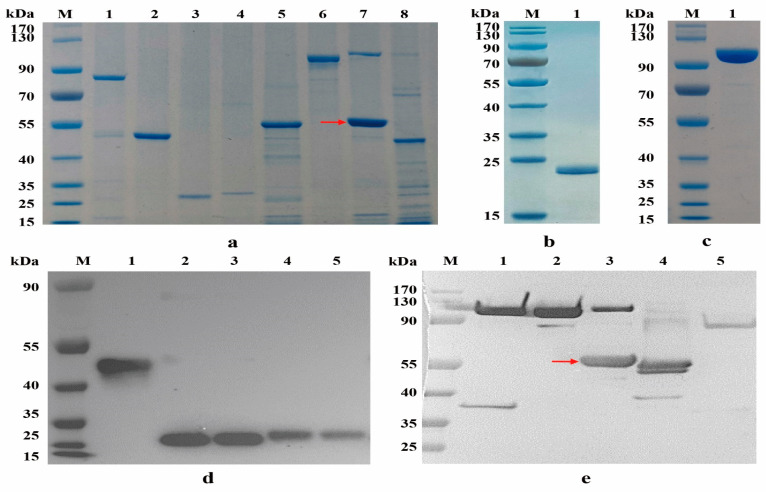
Identification of recombinant proteins by SDS-PAGE (**a**–**c**) and Western blotting (**d**,**e**). (**a**) Lane 1, FH; lane 2, FHc; lane 3, FHc-C; lane 4, FHc-N; lane 5, FHN; lane 6, FL-HN, Lane 7, FL (Red arrow), the above protein may be a dimer, Lane 8, FsL-HN; (**b**) Lane 1, Trx-rVAMP2 (23 kDa), in where rVAMP2 (aa30-92) was fused to thioredoxin (Trx) to produce a fusion protein Trx-rVAMP2; (**c**) Lane 1, FmL-HN; (**d**) Lane 1, FHc, lanes 2–3, FHc-C; lanes 4–5, FHc-N; (**e**), lane 1, FmL-HN; lane 2, FL-HN, lane 3, FL (Red arrow), the above protein may be a dimer, lane 4; FHN, lane 5, FH.

**Figure 2 toxins-15-00200-f002:**
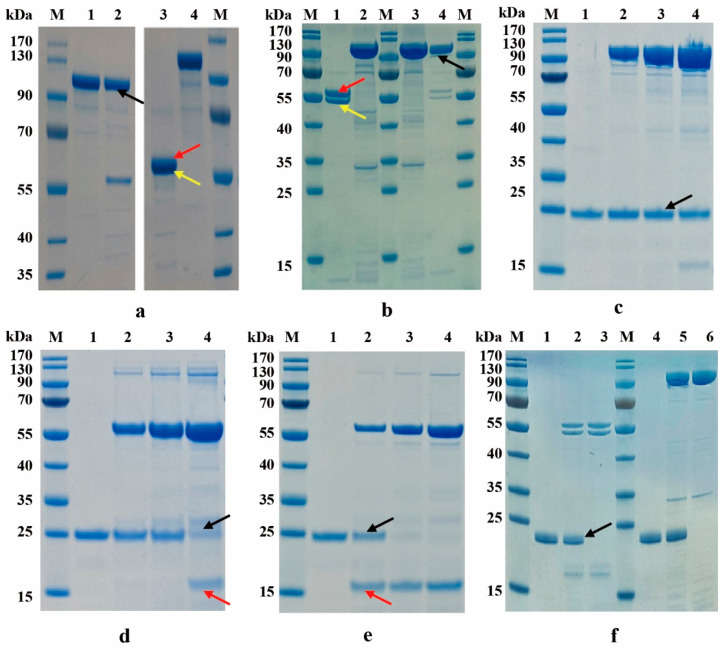
Functional activity and property of FL-HN. a and b, Protein structure analysis of FL-HN following nicking by trypsin. (**a**) Lanes 1 and 2, the un-nicked FL-HN (single chain, SC) and nicked FL-HN (dichain, DC) were determined by SDS-PAGE in the absence of the reductant, lanes 3 and 4, FL-HN-DC and FL-HN-SC were determined by SDS-PAGE in the presence of the reductant. The black arrow indicates the position of FL-HN; the red and yellow arrows represent FL and FHN, respectively. (**b**) Lanes 1 and 2, the FmL-HN-DC and FmL-HN-SC were determined by SDS-PAGE in the presence of reductant, lanes 3 and 4; FmL-HN-SC and FmL-HN-DC were determined by SDS-PAGE in the absence of the reductant. The red and yellow arrows represent FL and FHN, respectively. Black arrows indicated the position of FmL-HN. (**c**–**f**) Trx-rVAMP2 (23 kDa), rVAMP2 (aa 30–92) was fused to Trx to produce a fusion protein as substrate protein for BoNT/F. The protease activity of BoNT/F functional molecules to cleave rVAMP2 in vitro was determined by SDS-PAGE. (**c**–**e**) The results for FL-HN-SC, FL-HN-DC, and FL cleavage of rVAMP2, respectively. Lanes 1–4, the molar ratios of functional molecules to rVAMP2 protein were 0:1, 1:2, 1:1, and 2:1 from left to right. The 3 µmol/L concentration of rVAMP2 in each group was consistent. The black and red arrows indicate the uncleaved and cleaved rVAMP2, respectively. (**f**) Lane 1, rVAMP2 protein without FmL-HN-DC, lane 2, rVAMP2 protein with FmL-HN-DC, lane 3, FmL-HN-DC without rVAMP2 protein, lane 4, rVAMP2 protein without FmL-HN-SC, lane 5, rVAMP2 protein with FmL-HN-SC, lane 6, FmL-HN-DC without rVAMP2 protein. The molar ratio of FmL-HN-DC or FmL-HN-SC to rVAMP2 protein was 2:0. The black arrows indicated the uncleaved rVAMP2. These proteins above Trx-rVAMP2 were BoNT/F functional molecules.

**Figure 3 toxins-15-00200-f003:**
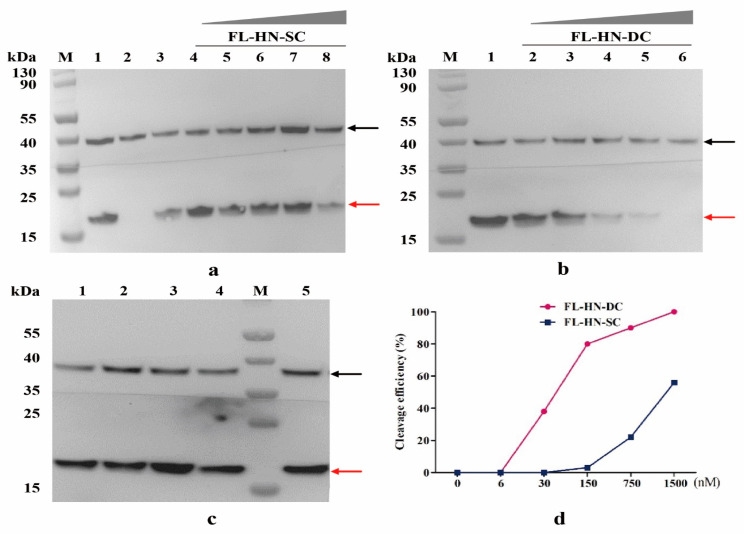
Functional activity and property of FL-HN in neuro-2a cell. (**a**–**c**) Protease activity of BoNT/F functional molecules cleaving VAMP2 substrates was determined using Western blotting in neuro-2a cells (one of three independent experiments). GAPDH was used as an internal reference. The black arrow indicates the GAPDH. The red arrow indicates the uncleaved VAMP2. (**a**) Lane 1, untreated cells, lane 2, 250 nM toxin, lane 3, 1500 nM FL, lane 4, 6 nM FL-HN-SC, lane 5, 30 nM FL-HN-SC, lane 6, 150 nM FL-HN-SC, lane 7, 750 nM FL-HN-SC, lane 8, 1500 nM FL-HN-SC; (**b**) lane 1, untreated cells, lane 2, 6 nM FL-HN-DC, lane 3, 30 nM FL-HN-DC, lane 4, 150 nM FL-HN-DC, lane 5, 750 nM FL-HN-DC, lane 6, 1500 nM FL-HN-DC; (**c**) lane 1, 500 nM FmL-HN-SC, lane 2, 1500 nM FmL-HN-SC, lane 3, 500 nM FmL-HN-DC, lane 4, 1500 nM FmL-HN-DC, lane 5, untreated cells. (**d**) Activity efficiency of FL-HN-DC and FL-HN-SC cleaving VAMP2 substrate protein in neuro-2a cells. Results were obtained by densitometry analysis and plotted. Densitometric analysis was performed using Image J software and nonlinear fitting was performed using GraphPad prism 8.

**Figure 4 toxins-15-00200-f004:**
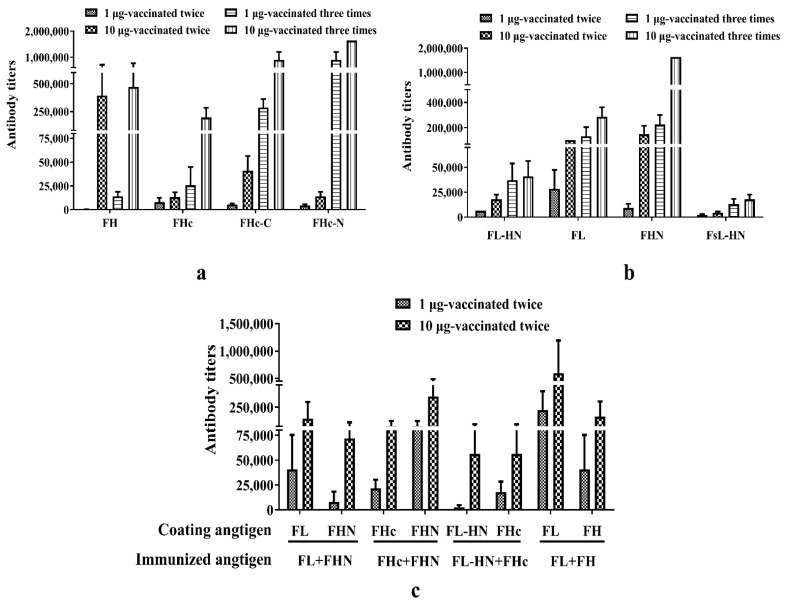
Sera antibody titers in mice after immunization with different antigens. Serum samples of the individual mice (*n* = 10, each group) were obtained after two or three immunizations and ELISA was used to measure the specific antibody titers. Individual mouse serum samples were assayed for each group, the geometric mean titer (GMT) ± standard deviation (SD) was calculated for this group. (**a**) anti-FH, FHc, FHc-C, and FHc-N antibody titers in mice immunized with the respective antigen molecule. Each antigen on the *x*-axis has four mouse immunization groups, two and three vaccinations with 1 µg dose antigen, two and three vaccinations with 10 µg dose antigen; (**b**) anti-FL-HN, FL, FHN, and FsL-HN antibody titers in mice immunized with the respective antigen molecule. Each antigen on the *x*-axis has four mouse immunization groups, two and three vaccinations with 1 µg dose antigen, two and three vaccinations with 10 µg dose antigen; (**c**) specific antibody titers in mice immunized with the combined antigens. The immunized combined antigens on the *x*-axis has two mouse immunization groups, two and three vaccinations with 1 µg dose antigen. Each antigen-specific antibody titers of combined antigen groups were detected with coating antigen on the *x*-axis. Here, single chain FL-HN (FL-HN-SC) was used to assess its humoral immune response.

**Table 1 toxins-15-00200-t001:** Protective ability and neutralization antibody titers in mice immunized with different functional molecule antigens.

Vaccine (Dosage)	Two Immunizations				Three Immunizations			
	Number of Survivors ^a^			NA (IU/mL) ^c^	Number of Survivors ^a^			NA (IU/mL) ^c^
	10^2 b^	10^3 b^	10^4 b^		10^2 b^	10^3 b^	10^4 b^	
1 µg FH	10/10	10/10	10/10	3.0	10/10	10/10	10/10	5.0
10 µg FH	10/10	10/10	10/10	12	10/10	10/10	10/10	15
1 µg FHc	10/10	10/10	10/10	2.0	10/10	10/10	10/10	40
10 µg FHc	10/10	10/10	10/10	8.0	10/10	10/10	10/10	80
1 µg FHc-C	2/10 *	ND	ND	<0.1	0/10 **	ND	ND	<0.1
10 µg FHc-C	0/10 **	ND	ND	<0.1	0/10 **	ND	ND	<0.1
1 µg FHc-N	2/10 *	0/10 **	ND	<0.1	8/10	0/10 **	ND	<0.1
10 µg FHc-N	8/10	0/10 **	ND	<0.1	8/10	0/10 **	ND	<0.1
1 µg FL-HN-SC	10/10	10/10	10/10	10	10/10	10/10	10/10	40
10 µg FL-HN-SC	10/10	10/10	10/10	20	10/10	10/10	10/10	60
1 µg FL	2/10 *	ND	ND	0.1	10/10	0/10 **	ND	0.4
10 µg FL	2/10 *	ND	ND	0.2	10/10	0/10 **	ND	0.8
1 µg FHN	2/10 *	ND	ND	<0.1	6/10	1/10 ^#^	ND	≤0.2
10 µg FHN	2/10 *	ND	ND	<0.1	8/10	2/10 *	ND	1.0
1 µg FsL-HN	0/10 **	ND	ND	<0.1	0/10 **	ND	ND	<0.1
10 µg FsL-HN	0/10 **	ND	ND	<0.1	0/10 **	ND	ND	≤0.2
PBS	0/10 **	ND	ND	<0.1	0/10 **	ND	ND	<0.1

^a^ The 10^2^, 10^3^, or 10^4^ × LD_50_ of BoNT/F was used to challenge the mice (*n* = 10/group) at 3 weeks after two or three immunizations (Number of lives/total). ^b^ The neurotoxin dose of BoNT/F (LD_50_). ^c^ The sera of the immunized mice and neurotoxin were mixed in vitro, and then the mice were injected intraperitoneally to detect the average neutralizing antibody (NA) titers in each group. ND means not determined. ** *p* = 0.00001, * *p* = 0.0007, # *p* = 0.0001, compared with the fully protected group (10/10) such as FH, FHc, or FL-HN antigen groups.

**Table 2 toxins-15-00200-t002:** Protective ability and neutralization antibody titers in mice immunized with different combined antigens.

Vaccine (Total Dosage)	Two Immunizations			
	Number of Survivors ^a^			NA (IU/ML) ^c^
	10^2 b^	10^3 b^	10^4 b^	
FL+FH (1 µg)	10/10	10/10	10/10	30
FL+FH (10 µg)	10/10	10/10	10/10	40
FL-HN-SC+FHc (1 µg)	10/10	10/10	10/10	40
FL-HN-SC+FHc (10 µg)	10/10	10/10	10/10	80
FL+FHN (1 µg)	10/10	10/10	10/10	2.0
FL+FHN (10 µg)	10/10	10/10	10/10	2.0
1 µg FL-HN-SC	10/10	10/10	10/10	10
10 µg FL-HN-SC	10/10	10/10	10/10	20
FHc+FHN (1 µg)	10/10	10/10	10/10	60
FHc+FHN (10 µg)	10/10	10/10	10/10	80
1 µg FH	10/10	10/10	10/10	3.0
10 µg FH	0/10 **	ND	ND	12
FHc-C+FHc-N (1 µg)	0/10 **	0/10 **	ND	<0.1
FHc-C+FHc-N (10 µg)	0/10 **	ND	ND	<0.2
1 µg FHc	10/10	10/10	10/10	2.0
10 µg FHc	10/10	10/10	10/10	8.0
PBS	0/10 **	ND	ND	<0.1

^a^ The 10^2^, 10^3^, or 10^4^ LD_50_ of BoNT/F was used to challenge the mice (*n* = 10/group) at 3 weeks after two or three immunizations (Number of lives/total). ^b^ The neurotoxin dose of BoNT/F. ^c^ The sera of the immunized mice and neurotoxin were mixed in vitro, and then the mice were injected intraperitoneally to detect the average neutralizing antibody (NA) titers in each group. ND means not determined. ** *p* = 0.00001, compared with the fully protected group (10/10).

**Table 3 toxins-15-00200-t003:** Protective ability and neutralization antibody titers in mice immunized of different molecular forms of FL-HN.

Vaccine (Dosage)	Two Immunizations				Three Immunizations			
	Number of Survivors ^a^			NA (IU/mL) ^c^	Number of Survivors ^a^			NA (IU/mL) ^c^
	10^2 b^	10^3 b^	10^4 b^		10^2 b^	10^3 b^	10^4 b^	
4 µg FL-HN-SC	10/10	10/10	10/10	30	10/10	10/10	10/10	80
1 µg FL-HN-SC	10/10	10/10	10/10	15	10/10	10/10	10/10	40
4 µg FmL-HN-SC	10/10	10/10	10/10	30	10/10	10/10	10/10	40
1 µg FmL-HN-SC	10/10	10/10	10/10	15	10/10	10/10	10/10	20
4 µg FL-HN-DC	10/10	10/10	10/10	10	10/10	10/10	10/10	40
1 µg FL-HN-DC	10/10	10/10	10/10	4.0	10/10	10/10	10/10	10
4 µg FmL-HN-DC	10/10	10/10	10/10	5.0	10/10	10/10	10/10	20
1 µg FmL-HN-DC	10/10	10/10	10/10	2.0	10/10	10/10	10/10	20
PBS	0/10 **	ND	ND	<0.1	0/10	ND	ND	<0.1

^a^ The 10^2^, 10^3^, or 10^4^ LD_50_ of BoNT/F was used to challenge the mice (*n* = 10/group) at 3 weeks after two or three immunizations (Number of lives/total). ^b^ The neurotoxin dose of BoNT/F. ^c^ The sera of the immunized mice and neurotoxin were mixed in vitro, and then the mice were injected intraperitoneally to detect the average neutralizing antibody (NA) titers in each group. ND means not determined. ** *p* = 0.00001, compared with the fully protected group (10/10).

**Table 4 toxins-15-00200-t004:** Protective ability and neutralizing antibody titer of FL-HN-SC, FHc, and FmL-HN-SC after dose-dependent immunizing mice.

Vaccine (Dosage)	One Immunizations		Two Immunizations			
	Number of Survivors ^a^		Number of Survivors ^a^			NA (IU/mL) ^c^
	10^2 b^	10^3 b^	10^2 b^	10^3 b^	10^4 b^	
4 µg FL-HN-SC	10/10	10/10	10/10	10/10	10/10	30
1 µg FL-HN-SC	10/10	10/10	10/10	10/10	10/10	15
250 ng FL-HN-SC	10/10	10/10 *	10/10	10/10	10/10	12
62.5 ng FL-HN-SC	10/10	10/10 **	10/10	10/10	10/10	6.0
15.6 ng FL-HN-SC	2/10	0/10	10/10	10/10	10/10	3.0
4 µg FHc	ND	10/10	10/10	10/10	10/10	6.0
1 µg FHc	ND	10/10	10/10	10/10	10/10	4.0
250 ng FHc	ND	4/10	10/10	10/10	10/10	3.0
62.5 ng FHc	ND	1/10	10/10	10/10	10/10	1.0
15.6 ng FHc	ND	0/10	10/10	10/10	10/10	0.5
4 µg FmL-HN-SC	10/10	10/10	10/10	10/10	10/10	30
1 µg FmL-HN-SC	10/10	10/10	10/10	10/10	10/10	15
250 ng FmL-HN-SC	4/10	2/10	10/10	10/10	10/10	8.0
62.5 ng FmL-HN-SC	0/10	0/10	10/10	10/10	10/10	4.0
15.6 ng FmL-HN-SC	0/10	0/10	10/10	10/10	10/10	2.0

^a^ The 10^2^, 10^3^, or 10^4^ LD_50_ of BoNT/F was used to challenge the mice (*n* = 10/group) at 3 weeks after two or three immunizations (Number of lives/total). ^b^ The neurotoxin dose of BoNT/F. ^c^ The sera of the immunized mice and neurotoxin were mixed in vitro, and then the mice were injected intraperitoneally to detect the average neutralizing antibody (NA) titers in each group. ND means not determined. * *p* = 0.011, ** *p* = 0.0001, compared with the same dosages of FHc or FmL-HN groups.

## Data Availability

The data presented in this study are available upon request from the corresponding author.

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
