# Peer review of "Biological and Immunological Characterization of a Functional L-HN Derivative of Botulinum Neurotoxin Serotype F"

_toxins, 2023, doi:10.3390/toxins15030200_

Round 1

Reviewer 1 Report

This is a generally well-written and important paper. One significant omission is the subserotype used for the sequence.

Other comments

what was the source of the anti-Fc horse sera?

line 48 delete "together"

line 79: change "immune protection efficacy" to "protection"

Line 85 change "have good immune protection efficacy" to  "provide good protection"

line 95 I don't think you need the word "Molecule" functional domain captures the meaning

Line 105 change "According to the functional" to "Based on the"

Line 128 do you mean dimer? bivalent is not the right word here. Maybe disulphide crosslinked chains

line 157 change cytological to cellular

line 163 what is a gray value? do you mean signal intensity?

Line 169 change cytological to "cell-based"

line 196 seems like" d number of vaccinations" is a footnote in the wrong place

line 233 change "resist attack by" to "prevent death from"

Line 354 change "might have more or stronger antibody epitopes" to "may be more immunogenic"

Lines 602, 608. what happened after a week?

Lines 645-657 Several sentences seem to be  repeated

In the references: numbering is in the Article titles.

Author Response

Dear reviewer,

We would like to thank the reviewer for careful and thorough reading of this manuscript and for the thoughtful comments and constructive suggestions, which help to improve the quality of this manuscript. Our detailed responses to the reviewer comments can be found as follows. The comments from the reviewer are in italics and black, and responses from the authors are in blue. Any revisions to the manuscript was marked up using the “Track Changes” function. In addition, another revised manuscript version without the Track Changes also submitted to the journal.

The answers and details on the point-by-point reply to these comments as follow:

General Comments and Suggestions:

This is a generally well-written and important paper. One significant omission is the sub serotype used for the sequence.

Reply: We appreciate the positive feedback from the reviewer. In revised supplementary files-Table S1 and Figure S1, we add to describe all sequences for the BoNT/F (Langeland strain, accession:X81714, amino acids 1 to 1278) functional fragments and rVAMP2 in Table S1.

Other comments

  • what was the source of the anti-FHc horse sera?

Reply: The anti-FHc horse sera is from our team, which was prepared as describe in References 58: Yu, Y.Z.; Zhang, S.M.; Ma, Y.; Zhu, H.Q.; Wang, W.B.; Du, Y.; Zhou, X.W.; Wang, R..; Wang, S.; Yu, W.Y.; Huang, P.T.; Sun, Z.W., Development and evaluation of candidate vaccine and antitoxin against botulinum neurotoxin serotype F. Clin Immunol. 2010, 137(2), :271-80. The anti-BoNT/F horse sera was from National Institutes of Food and Drug Control, Beijing, China.“anti-FHc (1:2000) [58] and anti-BoNT/F (1:1000) horse sera antibodies (from National Institutes of Food and Drug Control, Beijing, China)”was revised in manuscript text.

  • line 48 delete "together"

line 79: change "immune protection efficacy" to "protection"

Line 85 change "have good immune protection efficacy" to  "provide good protection"

line 95 I don't think you need the word "Molecule" functional domain captures the meaning

Line 105 change "According to the functional" to "Based on the"

Reply: These above corrections have been made in our revised manuscript as suggested.

  • Line 128 do you mean dimer? bivalent is not the right word here. Maybe disulphide crosslinked chains

Reply: We mean dimer and revise dipolymer into dimer.

  • line 163 what is a gray value? do you mean signal intensity?

Reply: We mean signal intensity and revise gray value into signal intensity.

  • line 157 change cytological to cellular

Line 169 change cytological to "cell-based"

line 196 seems like" d number of vaccinations" is a footnote in the wrong place

line 233 change "resist attack by" to "prevent death from"

Line 354 change "might have more or stronger antibody epitopes" to "may be more immunogenic"

Reply: These above corrections have been made in our revised manuscript as suggested.

  • Lines 602, 608. what happened after a week?

Reply: In the experiment of Immunization of mice and challenge with BoNT/F, the immunized mice were challenged with BoNT/F by i.p. injection, and the survival of the mice was recorded for one week. Then, euthanasia of animals was performed for the living animal. After the animal experiment, the body will be stored in the -20℃ refrigerator, there will be a special treatment company put away a unified treatment. These words were added in revised manuscript.

  • Lines 645-657 Several sentences seem to be repeated

Reply: This is one little mistake. These repeated sentences were deleted in revised manuscript.

  • In the references: numbering is in the Article titles.

Reply: In the references, the numbering in the Article titles was deleted in revised manuscript. All references of the manuscript are also checked.

Reviewer 2 Report

The manuscript “Biological and immunological characterization of a functional L-HN derivative of botulinum neurotoxin serotype F” by XYZ et al. Their findings can provide new strategies for developing vaccines and therapeutic antibodies against Botulinum neurotoxins (BoNT). The manuscript is not written well; several grammatical and typographical errors exist. After thoroughly reviewing I feel the manuscript needs to be extensively revision

Comments:

1.     Abstract section is not written properly; I suggest rewriting the conclusion to make it easy to understand.  

2.     I will suggest adding more about the mechanism of action of BoNT in the introduction section

3.     Please check that the acronyms throughout the manuscript should be correctly used, completely written at first mention, and not required for single-use of words.

4.     Please provide information on statistical analysis, software, the significance level, and your results at the end of the materials and methods section.

5.     The result and discussion section should be improved with more results descriptions. Further discussion in-depth about each item should be added, including why it decreased or increased.

6.     To support your work, add more findings from other studies.

7.     I will suggest adding a graphical abstract, which is the point of attraction for a reader.

Author Response

Dear reviewer,

We would like to thank the reviewer for careful and thorough reading of this manuscript and for the thoughtful comments and constructive suggestions, which help to improve the quality of this manuscript. Our detailed responses to the reviewer comments can be found as follows. The comments from the reviewer are in italics and black, and responses from the authors are in blue. Any revisions to the manuscript was marked up using the “Track Changes” function. In addition, another revised manuscript version without the Track Changes also submitted to the journal.

The answers and details on the point-by-point reply to these comments as follow:

General Comments and Suggestions:

The manuscript “Biological and immunological characterization of a functional L-HN derivative of botulinum neurotoxin serotype F” by XYZ et al. Their findings can provide new strategies for developing vaccines and therapeutic antibodies against Botulinum neurotoxins (BoNT). The manuscript is not written well; several grammatical and typographical errors exist. After thoroughly reviewing I feel the manuscript needs to be extensively revision.

Reply: The manuscript is well checked by a native English-speaking Ph.D. In revised manuscript, a lot of grammatical and typographical errors have been revised and the logical relationship between some sentences and paragraphs is also improved. We believed that the revised manuscript is read well.

The manuscript was extensively revised by us. In our revised manuscript, the Abstract, Introduction and Discussion parts have been extensively revised and improved. The Result and Materials and Methods parts have also been extensively improved. The Table and Figure, Supplement materials, References have also been extensively improved in our revised manuscript.

Comments:

  1. Abstract section is not written properly; I suggest rewriting the conclusion to make it easy to understand.  

Reply: The Abstract section has been extensively revised and improved in revised manuscript according to your suggest. In the abstract section, the logical relationship is also improved.

  1. I will suggest adding more about the mechanism of action of BoNT in the introduction section.

Reply: the Introduction section has been extensively revised and improved in revised manuscript according to your suggest. For example, the mechanism of action of BoNT was added and 6 new references is cited here. The last two paragraph of the Introduction section has also been extensively improved in our revised manuscript.

  1. 3. Please check that the acronyms throughout the manuscript should be correctly used, completely written at first mention, and not required for single-use of words.

Reply: we check and revise the acronyms throughout the manuscript.

  1. Please provide information on statistical analysis, software, the significance level, and your results at the end of the materials and methods section.

Reply: we provide detailed information on statistical analysis, software in Data analysis part of the materials and methods section. The significance level and some results were listed in the text and tables of revised manuscript.

  1. The result and discussion section should be improved with more results descriptions. Further discussion in-depth about each item should be added, including why it decreased or increased.

Reply: The text has been revised as suggested. In the result section, a lot of content are suppled. Especially, in the Legend of figures and tables of revised manuscript, a lot of detailed results were added.

    In the discussion section, a lot of new content are suppled and improved in revised manuscript. More results descriptions on our studies and other studies are added and they are discussed with data of manuscript. Especially, in the third paragraph of the discussion section, some contents on the L and/or HN domains of BoNTs are added. In the fifth paragraph of the discussion section, some contents on the translocation domains in BoNT/A and BoNT/B are added. Some contents on the L chain of each BoNT enter cells and cleave the SNARE proteins are deleted or the part of them are moved to the induction section.

  1. To support your work, add more findings from other studies.

Reply: some findings from other studies were added in revised manuscript as suggested. For example, new references 43, 44 and 58, etc., are cited here. Specifically, previous similar studies were performed with the serotypes BoNT/A and /B. In the present work, BoNT/F was used as model system. The results show that the fusion of two domain (FL-HN) has the highest propensity as a vaccine.

  1. I will suggest adding a graphical abstract, which is the point of attraction for a reader.

Reply: I am very grateful for your advice. While, a graphical abstract is not necessary for this journal. It can added if this journal needs it. Moreover, the abstract part should be read well following thoroughly improving.

Reviewer 3 Report

The manuscript describes the immunological characterization of several subunit based vaccine for BoNT/F. The authors demonstrate effective elicitation of neutralizing antibodies and protective response by using several constructs.

I have several comments:

-        -  The authors are not consistent with the way they name the constructs, which cause a confusion. For example, FL-HN is also described as FL-HN-SC etc.

-         -  Figure 4 – it is not indicated what are the bars represent? Average of how many duplicates? And what are the standard error?

-         - The authors conclude that the FL-HN-SC construct is better than FHc and the toxoid vaccine. They have provided the results of the FHc in the same model of vaccination and challenge, yet the results suggest that the differences are not significant (e.g. table 2 – 8 Vs. 20 IU/ml). In addition, the authors should add the results of the toxoid-based response and protection in this same model (even though they claim that this vaccine was not very effective in human trials). Only by comparing vaccines in the same model, one will be able to comment about their differential efficacy.

Author Response

Dear reviewer,

We would like to thank the reviewer for careful and thorough reading of this manuscript and for the thoughtful comments and constructive suggestions, which help to improve the quality of this manuscript. Our detailed responses to the reviewer comments can be found as follows. The comments from the reviewer are in italics and black, and responses from the authors are in blue. Any revisions to the manuscript was marked up using the “Track Changes” function. In addition, another revised manuscript version without the Track Changes also submitted to the journal.

The answers and details on the point-by-point reply to these comments as follow:

General Comments:

The manuscript describes the immunological characterization of several subunit based vaccine for BoNT/F. The authors demonstrate effective elicitation of neutralizing antibodies and protective response by using several constructs.

Reply: Thank you for your comment.

Several comments

  • The authors are not consistent with the way they name the constructs, which cause a confusion. For example, FL-HN is also described as FL-HN-SC etc.

Reply: As suggested by the reviewer, we have developed and defined SC and DC form on FL-HN and carefully revised FL-HN into FL-HN-SC. The two structure forms of FL-HN (i.e., FL-HN-SC: single chain FL-HN and FL-HN-DC: di-chain FL-HN) were developed and identified. In next experiments, FL-HN-SC or/and FL-HN-DC were used. FL-HN is described as FL-HN-SC when FL-HN was used as single chain, while FL-HN is described as FL-HN-DC when FL-HN was used as di-chain.

  • Figure 4 – it is not indicated what are the bars represent? Average of how many duplicates? And what are the standard error?

Reply: In Figure 4, these bars were marked to represent four immunization groups of each antigen. In addition, we improved the illustration of Figure 4 in revised manuscript. For example, in Figure 4a, each antigen (FH, FHc, FHc-C and FHc-N) on the x-axis has four mouse immunization groups, two and three vaccinations with 1 μg dose antigen, two and three vaccinations with 10 μg dose antigen; In Figure 4b, each antigen (FL-HN, FL, FHN and FsL-HN) on the x-axis has four mouse immunization groups, two and three vaccinations with 1 μg dose antigen, two and three vaccinations with 10 μg dose antigen; In Figure 4c, the immunized combined antigens (FL+FHN, FHc+FHN, FL-HN+FHc, FL+FH) on the x-axis has two mouse immunization groups, two vaccinations with 1 μg and 10 μg dose antigen. Each antigen-specific antibody titers of combined antigen groups were detected with coating antigen on the x-axis.

Serum samples of the individual mice (n = 10, each group) were obtained after two or three immunizations and ELISA was used to measure the specific antibody titers. In Figure 4, individual mouse serum samples (10 samples of per group) were assayed for each group, the geometric mean titer (GMT) ± standard deviation (SD) was calculated for this group. The SD is marked on the Bars of each group.

In addition, Fig.4 of high resolution was replaced in revised manuscript.

  • The authors conclude that the FL-HN-SC construct is better than FHc and the toxoid vaccine. They have provided the results of the FHc in the same model of vaccination and challenge, yet the results suggest that the differences are not significant (e.g. table 2 – 8 Vs. 20 IU/ml). In addition, the authors should add the results of the toxoid-based response and protection in this same model (even though they claim that this vaccine was not very effective in human trials). Only by comparing vaccines in the same model, one will be able to comment about their differential efficacy.

Reply: For first question, in this study, the FL-HN-SC construct is better protection than FHc vaccine. In tables 1 and 2, we have provided the results of the FHc and FL-HN in the same model of vaccination and challenge, the differences are not significant. While FL-HN induced a neutralizing titer that was 2~3 times higher than that of the FHc group (10 IU/ml VS 2 IU/ml at the 1 μg dose; 20 IU/ml VS 8 IU/ml at the 10 μg dose).

However, in table 4, FL-HN, FmL-HN, and FHc showed a dose-dependent response to BoNT/F after the first immunization. The protection rate of FL-HN with 62.5 ng in each mouse against 103 × LD50 of BoNT/F challenge was 100%, FmL-HN and FHc could protect mice completely against 103 × LD50 of BoNT/F challenge when the immunization dosage was 1 μg. The median effective protective dose (ED50) of FL-HN was 43.9 ng, and those of FmL-HN and FHc were 258.9 ng and 200 ng, respectively. The differences are significant, which also were added in table 4. FL‑HN, FmL-HN, and FHc did not show a significant difference in protective effect after two immunizations, while FL-HN had a stronger neutralizing ability, and the efficacy value detected at the same dose was 2~6 times that of FHc. These results indicated that FL-HN was a more effective target antigen for BoNT/F than FHc.

For second question, in our paper we describe that recombinant Hc or L-HN subunit vaccines have significant advantages compared with the toxoid vaccines. FL-HN-SC was the best immunogen for BoNT/F and could be used as a subunit vaccine to replace the FHc subunit vaccine and toxoid vaccine. In our paper, we do not conclude that the FL-HN-SC construct is better protection than the BoNT/F toxoid vaccine. In fact, we also approve that this is not concluded if there is not comparing vaccines in the same model. Now, it is regretful that no BoNT/F toxoid antigen or BoNT/F toxoid vaccine can be used to immunize animals. Therefore, we also do not complete this experiment of comparing the efficacy between FL-HN and BoNT/F toxoid.

Round 2

Reviewer 2 Report

The revised version of the manuscript has improved.

Author Response

Dear reviewer,

We would like to thank the reviewer for careful and thorough reading of this manuscript. Our detailed responses to the reviewer comments can be found as follows.

General Comments: English language and style are fine/minor spell check required.

Reply: The manuscript is well checked by us and a lot of grammatical and typographical errors have been revised in our revised manuscript. Any revisions to the manuscript was marked up using the “Track Changes” function.
